# Biocompatibility and Biological Effects of Surface-Modified Conjugated Polymer Nanoparticles

**DOI:** 10.3390/molecules28052034

**Published:** 2023-02-21

**Authors:** Wanni Guo, Mingjian Chen, Yuxin Yang, Guili Ge, Le Tang, Shuyi He, Zhaoyang Zeng, Xiaoling Li, Guiyuan Li, Wei Xiong, Steven Wu

**Affiliations:** 1NHC Key Laboratory of Carcinogenesis and Human Key Laboratory of Cancer Metabolism, Hunan Cancer Hospital and the Affiliated Cancer Hospital of Xiangya School of Medicine, Central South University, Changsha 410000, China; 2Key Laboratory of Carcinogenesis and Cancer Invasion of the Chinese Ministry of Education, Cancer Research Institute and School of Basic Medicine Sciences, Central South University, Changsha 410000, China; 3Hunan Key Laboratory of Nonresolving Inflammation and Cancer, Disease Genome Research Center, the Third Xiangya Hospital, Central South University, Changsha 410000, China; 4Department of Chemistry, University of South Dakota, Vermillion, SD 57069, USA

**Keywords:** surface modification, semiconducting polymer nanoparticles, biocompatibility

## Abstract

Semiconductiong polymer nanoparticles (Pdots) have a wide range of applications in biomedical fields including biomolecular probes, tumor imaging, and therapy. However, there are few systematic studies on the biological effects and biocompatibility of Pdots in vitro and in vivo. The physicochemical properties of Pdots, such as surface modification, are very important in biomedical applications. Focusing on the central issue of the biological effects of Pdots, we systematically investigated the biological effects and biocompatibility of Pdots with different surface modifications and revealed the interactions between Pdots and organisms at the cellular and animal levels. The surfaces of Pdots were modified with different functional groups, including thiol, carboxyl, and amino groups, named Pdots@SH, Pdots@COOH, and Pdots@NH_2_, respectively. Extracellular studies showed that the modification of sulfhydryl, carboxyl, and amino groups had no significant effect on the physicochemical properties of Pdots, except that the amino modification affected the stability of Pdots to a certain extent. At the cellular level, Pdots@NH_2_ reduced cellular uptake capacity and increased cytotoxicity due to their instability in solution. At the in vivo level, the body circulation and metabolic clearance of Pdots@SH and Pdots@COOH were superior to those of Pdots@NH_2_. The four kinds of Pdots had no obvious effect on the blood indexes of mice and histopathological lesions in the main tissues and organs. This study provides important data for the biological effects and safety assessment of Pdots with different surface modifications, which pave the way for their potential biomedical applications.

## 1. Introduction

In recent years, nanomaterials with size of 1–100 nm have developed rapidly in the field of biomedicine [1]. There are many types of nanoparticles, including quantum dots (QDs) [2,3,4], silicon nanoparticles [5,6], metal nanoclusters [7], etc. Quantum dots are a semiconductor material [8] whose internal electrons are restricted in all directions, resulting in a discontinuous electronic-energy-level structure. Quantum dots demonstrate characteristics of wide absorption range, narrow emission peak width, and excellent photostability and brightness [9]. However, the inherent heavy metal toxicity and complex preparation methods of quantum dots greatly limit the further application of these nanomaterials in in vivo imaging and clinical translation. To solve these limitations, various types of nanomaterials were developed, including semiconducting polymer dots (Pdots). Pdots are nanoparticles formed with semiconducting polymers by methods such as nanoprecipitation or microemulsion. Semiconducting polymers are macromolecular compounds with large π-conjugated backbones and delocalized electronic structures [10]. Since Pdots have a conjugated framework and a large number of π-π bonds, Pdots have strong light-harvesting ability, high fluorescence quantum yield, and easy electron and energy transfer in the conjugated framework [11]. Pdots also have the characteristics of excellent photostability, structural diversity, functional designability, and easy surface modification [12]. These outstanding properties ensured that Pdots could be widely used in the biomedical field, including bioimaging, biosensors, tumor therapy, etc. For instance, Day et al. synthesized a multimodal combination of Pdots containing a heat shock protein inhibitor and azo compound for triple-negative breast cancer [13].Wu et al. designed a silica nanoparticle coated with catalase to treat glioblastoma through chemodynamic therapy and sound dynamic therapy [6]. Deng et al. synthesized a manganese-loaded fluoride nanoparticle for MRI and chemodynamic treatment of tumors [14]. Özenler et al. developed Pdots for the cellular imaging and detection of cancer cells. Compared with normal hepatocytes, these Pdots could target the nuclear region of cancer cells, and the enhanced fluorescent brightness of the Pdots could be used to distinguish cancer cells from healthy hepatocytes in co-culture. This strategy provided an important means for the histological diagnosis of cancer cells [15]. Moreover, Lyu et al. applied Pdots for photoacoustic and photothermal therapy [16]. The semiconducting polymers and ultra-small carbon dots in this probe induced photoelectron transfer under light irradiation, which greatly improved the photothermal conversion efficiency and enhanced the photoacoustic signal intensity. In vitro and in vivo experiments demonstrated the probes performed photoacoustic imaging and photothermal therapy of tumors.

With the development of biomedical applications of Pdots, their biological effects and biosafety in cells and in vivo lack systematic investigation and summary. This severely impedes the further biomedical application and clinical translation of Pdots. In recent years, research on biocompatibility has been carried out accordingly. At the cellular level, Fernando et al. studied the effect of Pdots on the cytotoxicity and cell proliferation of J774A1 cells, and the results showed no effect in the investigated concentration range [17]. Ye et al. compared the effects of Pdots and QDs on cytotoxicity and on cellular oxidative stress. The results showed that Pdots were significantly less cytotoxic than QDs. At the same concentration, the fluorescence intensity of Pdots is nearly ten times higher than that of QDs [18]. At the animal level, Wu et al. investigated the biocompatibility of Pdots in pregnant mice and embryos. The results showed that injection of CPNs had no effect on the physiological and biochemical indicators of pregnant mice and was safe for the reproductive ability of mice [19]. Although these studies have examined the biological effects and biosafety of Pdots to a certain extent, no systematic investigation of the impact of surface modification on the toxicity of Pdots has been conducted.

In this study, we synthesized Pdots using poly(2-methoxy-5-(2-ethylhexyloxy)-1,4-phenylenevinylene) (MEH-PPV) and modified four different surface functional groups, including bare surface, thiol group, carboxyl, and amino groups, respectively. They were named Pdots, Pdots@SH, Pdots@COOH, and Pdots@NH_2_, respectively. We evaluated the effect of different surface modifications on the in vitro biological effects of nanoparticles by characterization and cellular experiments. At the animal level, we investigated the effects of several modifications on biodistribution and biocompatibility through in vivo imaging and toxicological analysis. This study contributed to the understanding of the effects of different surface modifications on the biotoxicity and biodistribution of Pdots, which would promote the application and translation to clinical settings of nanomaterials.

## 2. Results and Discussion

### 2.1. Synthesis and Characterization of Pdots, Pdots@SH, Pdots@COOH, and Pdots@NH_2_

The synthesis and surface modification of Pdots are shown in Figure 1. The semiconducting polymer (MEH-PPV) and amphiphilic polymer DSPE-PEG-R (0, -SH, -COOH, and -NH_2_) were used to synthesize Pdots, Pdots@SH, Pdots@COOH, and Pdots@NH_2_ through the nanoprecipitation method. From the TEM images, the nanoparticles are spherical and uniformly dispersed (Figure 2A). The sizes of Pdots, Pdots@SH, Pdots@COOH, and Pdots@NH_2_ were all 50 ± 20 nm. Hydration particle sizes were also measured (Figure 2B) and the results were consistent with those shown by TEM. These results indicated that three modifications of thiol, carboxyl, and amino groups had almost no effect on the morphology and size of Pdots. In addition, we recorded the UV-Vis-NIR absorption spectra and fluorescence spectra of the four nanoparticles by UV spectrophotometers and fluorescence spectrophotometers (Figure 2C,D). The results demonstrated that surface modification had an impact on the absorption and fluorescence spectra of Pdots. All of them showed an absorption peak at 480 nm and emission peak at 590 nm.

Figure 2E showed the zeta potentials of the four nanoparticles. The zeta potentials of Pdots, Pdots@SH, and Pdots@COOH are −16.3 mV, −18 mV, and −18 mV, respectively. In contrast, the zeta potential of Pdots@NH_2_ is +3.73 mV. Amino modification reversed the surface zeta potential of Pdots. Zeta potential is a predictor of stability in particulate dispersions to some extent. The higher the absolute value, the greater the electrostatic repulsion between the particles in the fluid. Therefore, the higher absolute zeta potential value indicated a better stability [20]. The absolute value of the zeta potential of Pdots@NH_2_ is only 3.73 mV among these four nanoparticles, which indicates that its stability was worse than that of Pdots, Pdots@SH, and Pdots@COOH. To further verify the stability of Pdots, Pdots@SH, Pdots@COOH, and Pdots@NH_2_, we measured the polymer dispersion index (PDI) of these four nanoparticles (Figure 2F). The PDI value of Pdots@NH_2_ was higher than that of the other three nanoparticles. The greater the PDI value, the worse the stability. Finally, we further detected the PDI of the four nanoparticles over the 7 days. The results are shown in Figure 2G. The PDI of these four nanoparticles increased slowly with time. Meanwhile, the PDI of Pdots@NH_2_ was significantly higher than that of the others at the same time point.

### 2.2. In Vitro Assessment of Biocompatibility

At the cellular level, we explored the biocompatibility of Pdots with different surface modifications. To investigate the toxic side effects of Pdots, Pdots@SH, Pdots@COOH, and Pdots@NH_2_ on cells, we incubated the four types of nanoparticles with human cervical cancer cells (CaSki), mouse breast cancer cells (4T1), and human normal lung epithelial cells (BEAS-2B) cells, respectively, for 24 h. We used CCK-8 assay to detect cell viability (Figure 3A–D). The results showed that in the three cell lines, the cell viability after incubation with the four nanoparticles slowly decreased with the increase in the concentration. Pdots@NH_2_ was more cytotoxic than the other three nanoparticles. When the concentration of Pdots@NH_2_ was 60 μg/mL, the cellular survival rate decreased to 80%. With the same concentration, the cells treated with the other three nanoparticles were all about 90%. Compared with Pdots, Pdots@SH and Pdots@COOH reduced the cytotoxicity to a certain extent.

To further verify the above results, we used flow cytometry to detect the apoptosis of cells after incubating with the four nanoparticles. The results showed that the cell viability of Pdots@SH and Pdots@COOH was slightly better than that of the Pdots group, while the cell viability of Pdots@NH_2_ was lower than that of the Pdots group (Figure 3E). This was consistent with the results of the CCK-8 assay. Overall, Pdots@SH and Pdots@COOH showed less cytotoxicity than Pdots. However, the modification of amino groups increased the toxicity, which might be caused by the easy aggregation.

Due to the high fluorescence properties of the MEH-PPV contained in Pdots, we studied the cellular uptake behavior of different surface-modified Pdots using fluorescence imaging. Based on the results of the CCK-8 assay, we selected a concentration of 50 µg/mL for subsequent cell experiments. The nuclei were stained with DAPI and showed blue fluorescence under a fluorescence microscope. Fluorescence imaging showed that four nanoparticles could enter CaSki cells efficiently after 6 h (Figure 4A), with the highest fluorescence intensity with Pdots@SH (Figure 4B). This might be caused by the thiol-disulfide exchange reaction, in which thiol groups on Pdots could interact with thiols on the side of cell membrane proteins, enhancing their cellular uptake [21]. In contrast, amino-modified nanoparticles showed the lowest fluorescence intensity in cells, indicating lower cell uptake. This may be due to the low absolute value of the zeta potential (+3.78 mV) with Pdots@NH_2_, which easily leads to particle aggregation [22]. In addition, we examined the oxidative stress of CaSki cell lines after exposure to Pdots, Pdots@SH, Pdots@COOH, and Pdots@NH_2_. The levels of reactive oxygen species (ROS) in cells were measured using DCFH-DA, an ROS probe. As shown in Figure 4C, the positive control showed bright green fluorescence, while cells treated with Pdots (a), Pdots@SH (b), Pdots@COOH (c), and Pdots@NH_2_ (d) showed no obvious green fluorescence, indicating that none of the four nanoparticles produced significant ROS. In summary, Pdots@SH and Pdots@COOH showed better cellular uptake and less toxicity compared with Pdots. However, Pdots@NH_2_ had poorer cellular uptake and increased cytotoxicity.

### 2.3. In Vivo Imaging and Biodistribution

To investigate the biodistribution of Pdots (a), Pdots@SH (b), Pdots@COOH (c), and Pdots@NH_2_ (d) in vivo, we injected Pdots (a), Pdots@SH (b), Pdots@COOH (c), and Pdots@NH_2_ (d) into different groups of mice via the caudal vein and performed in vivo fluorescence imaging at different times (1, 7, 15, 30, and 60 days). As shown in Figure 5A, all nanoparticles were circulated through the whole body through the blood, and most of the fluorescent signals were observed in the mice on day 1. Pdots@NH_2_ has a positive charge on its surface, leading to the adsorption and potential aggregation of proteins in the body. The transport and distribution of these nanoparticles in vivo are affected by their aggregates [23], and thus their circulating ability in vivo is worse than that of the other three nanoparticles. The fluorescence intensity in the mice slowly weakened over time. When the time reached 60 days, the fluorescence signal in the mice had mostly disappeared, indicating that most of the nanoparticles with different surface modifications had been metabolized out of the body.

Next, we performed organ imaging of the heart, liver, spleen, lungs, and kidneys to further assess the biological distribution of these nanoparticles in mice. Fluorescence images of these five organs were shown in Figure 5B. One day after the tail vein injection of nanoparticles, the liver, kidneys, and lungs showed strong fluorescence signals. The fluorescence signal of the liver was the strongest, indicating that the nanoparticles accumulated in the liver in large quantities. The fluorescence signal of each organ gradually decreased over time, which was consistent with the results in Figure 5A. Studies have shown that the excretion of nanoparticles is mainly through the kidneys and liver. The upper limit of glomerular filtration particle size is 8 nm [20], and particles larger than 8 nm are not excreted through the kidneys, but through the liver and gallbladder [21]. Pdots@SH showed the strongest fluorescence intensity in the liver and the fastest metabolism, followed by Pdots@COOH, Pdots, and Pdots@NH_2_. Importantly, Pdots@NH_2_ accumulated in organs was more difficult to metabolize and still had a strong fluorescence signal after 60 days in the liver and kidneys. Due to the poor stability and aggregation of Pdots@NH_2_, the adsorption and metabolism of Pdots@NH_2_ in various tissues and organs were affected [24]. In summary, among the four nanoparticles, Pdots@SH and Pdots@COOH were superior to Pdots@NH_2_ in both in vivo circulation and scavenging metabolism.

### 2.4. In Vivo Toxicology Study

We systematically investigated the in vivo toxicity of nanoparticles with different surface modifications. After Pdots, Pdots@SH, Pdots@COOH, and Pdots@NH_2_ were injected into Balb/c mice at a dose of 10 mg/kg, their behavioral activities, body weight, etc., were monitored (Appendix A). The behavior of the mice did not show abnormalities, and there was no death. Their weight was stable and did not differ significantly from that of the control group. In addition, we also performed whole blood analysis and serum biochemical analysis on the orbital blood of mice at 1, 7, 15, 30, and 60 days after the intravenous injection of nanoparticles. The routine blood tests included the analysis for white blood cell (WBC), red blood cell (RBC), and platelet count (PLT), mean corpuscular volume (MCV), mean corpuscular concentration (MCHC), hemoglobin (HGB), and hematocrit (HCT). The results in Figure 6 showed that there were no significant differences between the mice treated with Pdots, Pdots@SH, Pdots@COOH, and Pdots@NH_2_ compared with the mice in the saline-injected group. The number and morphology of red blood cells, white blood cells, and platelets in whole blood were within normal limits. This indicated that Pdots with different surface modifications pose no significant risk to the blood routine indexes of mice. Next, serum biochemical analysis was performed. We mainly detected related liver function indicators (phosphatase (ALP), alanine aminotransferase (ALT), aspartate aminotransferase (AST), and total basic bilirubin (TBIL)) and renal function indicators (blood urea nitrogen (BUN) and creatinine (CRE)). As shown in Figure 7, these indicators of liver and kidney function did not change significantly compared with the control group. All parameters are within the normal reference range. Even though most of the four nanoparticles accumulated in the liver and kidneys, they had no obvious toxicity to the liver and kidneys, which did not affect the liver and kidney functions.

Pdots, Pdots@SH, Pdots@COOH, and Pdots@NH_2_ were injected into mice through the tail vein and reached organs through blood circulation. Therefore, we performed histopathological examination on the main organ tissues (heart, liver, spleen, lung, kidney) by H&E staining to determine whether these organs and tissues were damaged. The results showed that no obvious histopathological changes were observed in the heart, liver, spleen, lung, or kidneys due to the four kinds of nanoparticles after 1, 7, 15, and 60 days after entering the mice (Figure 8, Appendix A). These results indicated that Pdots, Pdots@SH, Pdots@COOH, and Pdots@NH_2_ had no significant toxic effects on mice.

## 3. Materials & Methods

### 3.1. Materials

Poly(2-methoxy-5-(2-ethylhexyloxy)-1,4-phenylenevinylene) (MEH-PPV), tetrahydrofuran (THF), 2′, 7′-Dichlorofluorescin diacetate (DCFH-DA), and 2-(4-Amidinophenyl)-6-indolecarba- midine dihydrochloride (DAPI) were obtained from Sigma-Aldrich (St. Louis, MO, USA). RPMI 1640 medium, fetal bovine serum (FBS), penicillin/treptomycin, paraformaldehyde, and trypsin were all purchased from Gibco Life Technologies Co., Ltd. (Grand Island, NY, USA). The Cell Counting Kit-8 (CCK-8) was provided by Beyotime Institute of Biotechnology (Haimen, Jiangsu, China). DSPE-PEG, DSPE-PEG-SH, DSPE-PEG-COOH, and DSPE-PEG-NH_2_ were obtained by Ruixi Biotech Co., Ltd. (Xi’an, Shanxi, China). Annexin V-FITC/PI Apoptosis Detection Kit was purchased from Yeasen Biotech Co., Ltd. (Shanghai, China).

### 3.2. Synthesis of Pdots, Pdots@SH, Pdots@COOH, and Pdots@NH_2_

Pdots, Pdots@SH, Pdots@COOH, and Pdots@NH_2_ were prepared by nanoprecipitation. A total of 5 mg MEH-PPV was dissolved in 10 mL tetrahydrofuran (THF) to obtain a 0.5 mg/mL MEH-PPV solution. A total of 10 mg DSPE-PEG was dissolved in 10 mL THF to obtain a 1 mg/mL DSPE-PEG solution. A total of 1 mL of 0.5 mg/mL MEH-PPV solution was mixed with 250 μL of 1 mg/mL DSPE-PEG solution in 3.5 mL of THF. The mixed solution was quickly poured into 10 mL of water under sonication. After five minutes, the THF in the solution was removed by a rotary evaporator. The resulting solutions were named Pdots. The concentration of Pdots was detected with a UV-Vis spectrometer. The preparation of the other three nanoparticles was similar to this procedure, replacing DSPE-PEG with DSPE-PEG-R (R=SH, COOH, NH_2_). The resulting nanoparticles were named Pdots@SH, Pdots@COOH, and Pdots@NH_2_, respectively.

### 3.3. Characterizations

TEM images were collected by Tecnai G2 F20 transmission electron microscopy (FEI, state abbreviation, USA). Hydrodynamic diameter, polymer dispersion index (PDI), and zeta potential were measured using a Zetasizer Nano particle sizer (Malvern Instruments, Malvern, UK). UV-vis spectra were measured by UV-2600 spectrophotometer (Hitachi, Tokyo, Japan). The fluorescence spectra were obtained by RF-6000 spectrofluorometer (Hitachi, Tokyo, Japan). Fluorescence images were obtained using an inverted fluorescence microscope (Zeiss, Oberkochen, Germany).

### 3.4. Cell Culture

Human cervical cancer cells (CaSki cells), mouse breast cancer cells (4T1 cells), and human normal lung epithelial cells (BEAS-2B cells) were all obtained from the Cancer Research institute, Central South University, Hunan, China. CaSki cells and 4T1 cells were cultured in RPMI-1640 medium with 10% FBS and 1% penicillin-streptomycin solution. BEAS-2B cells were cultured in RPMI-1640 medium with 10% FBS. All the cells were preserved in a humid atmosphere of 37 °C, containing 5% carbon dioxide.

### 3.5. In Vitro Cytotoxicity Assay

CaSki cells, 4T1 cells, and BEAS-2B cells were seeded into 96-well plates at a density of 8000 cells per well for 24 h and then incubated with different concentrations of Pdots, Pdots@SH, Pdots@COOH, and Pdots@NH_2_ for 24 h, and then CCK-8 solution was added. Cell viability was detected after 2 h of solution reaction.

### 3.6. Cell Apoptosis Assay

CaSki cells were seeded onto 6-well plates at 5 × 10^5^ cells per well for 24 h and incubated with 50 μg/mL of Pdots, Pdots@SH, Pdots@COOH, and Pdots@NH_2_ for 6 h, respectively. Cells were digested with trypsin (EDTA absent), washed twice with pre-cooled PBS, stained with 5 μL of V-FITC solution and 5 μL of PI solution for 15 min, and the cells were analyzed by flow cytometer.

### 3.7. In Vitro Cell Uptake

A total of 10^5^ CaSki cells were seeded in 24-well plates and incubated with 50 μg/mL of Pdots, Pdots@SH, Pdots@COOH, and Pdots@NH_2_ for 0 and 6 h, respectively. The medium was then removed, cells were fixed with 4% paraformaldehyde for 30 min, nuclei were stained with DAPI for 30 min and protected from light during the reaction. The cells were finally imaged with a fluorescence microscope.

### 3.8. Intracellular Reactive Oxygen Species (ROS) Detection

A total of 10^5^ CaSki cells were seeded in 24-well plates for 24 h. They were then incubated with medium containing 50 μg/mL of Pdots, Pdots@SH, Pdots@COOH, and Pdots@NH_2_ for 6 h. Cells were washed with PBS then stained with DCFH-DA for 30 min, and nuclei were stained by fixing cells with 4% paraformaldehyde for 30 min and incubating with DAPI for 30 min. Finally, the ROS signal was observed under a fluorescence microscope.

### 3.9. Animals

Healthy female BALB/c nude mice (4 weeks old) were purchased from Hunan SJA Laboratory Animal Co., Ltd. (Changsha, China). All procedures for animal experiments in this study were approved by the regulations of the Animal Experiment Ethics Committee of Central South University and were performed in accordance with the Institutional Animal Care and Use Committee of Central South University. Mice were housed in the sterile animal room of the Animal Experiment Department of Central South University, and the mice were acclimated to the animal facility for 7 days before the experiment.

### 3.10. In Vivo Imaging and Biodistribution

BALB/c nude mice were randomly divided into 4 groups: Group 1 was injected with Pdots solution (10 mg/kg) into the tail vein; Group 2 was injected with Pdots@SH solution (10 mg/kg) into the tail vein; Group 3 was injected with Pdots@COOH solution (10 mg/kg) into the tail vein; Group 4 was injected with Pdots@NH_2_ solution (10 mg/kg) into the tail vein. Then, on the 1st, 7th, 15th, 30th, and 60th days, the mice were imaged in vivo by a small animal imager to observe the distribution of nanoparticles in the mice. The mice were sacrificed by cervical dislocation, and the main organs (heart, liver, spleen, lung, and kidney) were imaged to observe the distribution of nanoparticles in tissues and organs.

### 3.11. Animal Treatments

Mice were randomly divided into 5 groups, as follows: Group 1 was injected with Pdots solution (10 mg/kg) into the tail vein; Group 2 was injected with Pdots@SH solution (10 mg/kg) into the tail vein; Group 3 was injected with Pdots@COOH solution (10 mg/kg) into the tail vein; Group 4 was injected with the Pdots@NH_2_ solution (10 mg/kg) into the tail vein; Group 5 was injected with saline into the tail vein. The behavioral activity status of the mice was observed, and the body weight of the mice was recorded every two days.

### 3.12. Hematology Analysis and Blood Biochemical Assay

At 1, 7, 15, 30, and 60 days after injection of mice with Pdots, Pdots@SH, Pdots@COOH, Pdots@NH_2_, and saline, 1 mL of blood was collected from each mouse through the orbital venous plexus into anticoagulant tubes. The blood routine was determined by an automatic hematology analyzer. The following hematological indicators were mainly selected: white blood cell (WBC) count, red blood cell (RBC) count, platelet (PLT) count, mean corpuscular volume (MCV), mean corpuscular concentration (MCHC), mean corpuscular hemoglobin (MCH), hemoglobin concentration (HGB), and hematocrit (HCT). A total of 1 mL of blood was collected from each mouse through the orbital venous plexus and centrifuged at 3000 rpm to obtain serum. In serum biochemical analysis, liver function and renal function indexes were mainly detected. Among them, the following liver function indexes were mainly detected: alanine aminotransferase (ALT), aspartate aminotransferase (AST), alanine aminotransferase (ALT), and total bilirubin (TBIL). Kidney function indicators included blood urea nitrogen (BUN) and creatinine (Cre). These analyses were performed with a veterinary blood analyzer.

### 3.13. Histopathological Examination

The major organs (heart, liver, spleen, lungs, and kidneys) were removed and fixed with 4% paraformaldehyde solution. The alcohol was dehydrated in ascending order from low to high concentration. After dehydration, the tissues were embedded in paraffin wax and sectioned. Tissue sections were dewaxed step by step from high to low alcohol concentration and stained with hematoxylin and eosin dyes. After staining, secondary dehydration and transparency were performed, and the section was sealed by drops of glue on the cover glass.

## 4. Conclusions

In conclusion, we investigated the biocompatibility and biodistribution of thiol-, carboxyl-, and amino-modified Pdots at the cellular and living animal levels. In vitro studies showed that the cellular uptake of thiol- and carboxyl-modified Pdots was strong and had no significant impact on cell viability, apoptosis, or oxidative stress. Amino modification decreased the uptake of Pdots and increased the cytotoxicity of Pdots to some extent. Next, we conducted an in vivo study and found that the nanoparticles were distributed in major tissues and organs of mice after injection, with the largest distribution in the liver. As time went by, the nanoparticles were eliminated from the mice by metabolism through the digestive system. Moreover, the in vivo circulation and metabolic clearance of Pdots@SH and Pdots@COOH were better than those of Pdots@NH_2_. The blood routine analysis, serum biochemical analysis, and histopathological analysis showed that Pdots modified with different modifications had no obvious toxicity and did not affect liver and kidney functions. The study demonstrated that thiol- and carboxyl-modified Pdots have better biocompatibility than amino-modified Pdots, which lays a solid foundation for the biomedical application and clinical translation of Pdots.

## Figures and Tables

**Figure 1 molecules-28-02034-f001:**
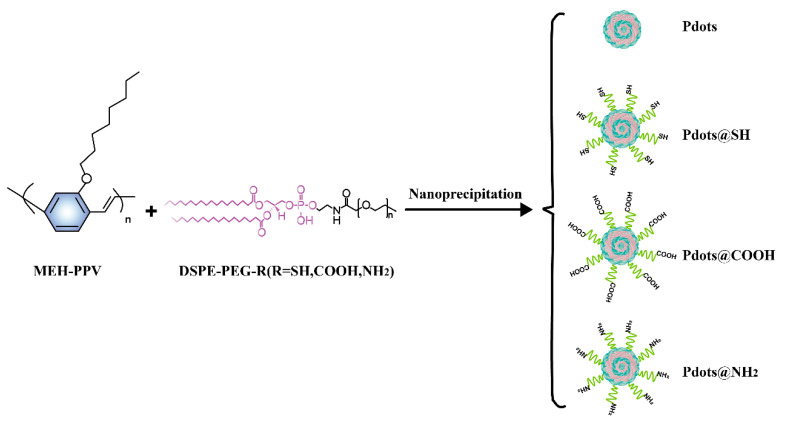
Scheme for the synthesis of Pdots, Pdots@SH, Pdots@COOH, and Pdots@NH_2_.

**Figure 2 molecules-28-02034-f002:**
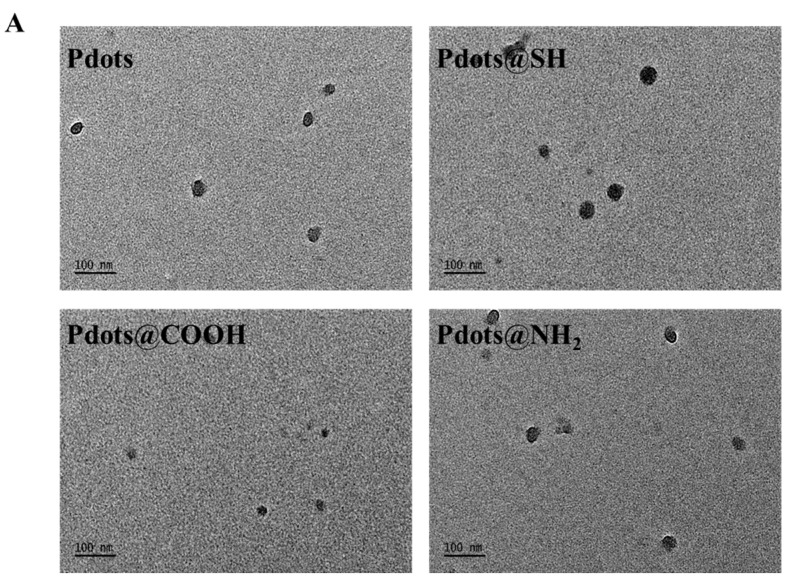
Characterization of Pdots, Pdots@SH, Pdots@COOH, and Pdots@NH_2_. (**A**) Representative transmission electron micrograph (TEM) images of Pdots, Pdots@SH, Pdots@COOH, and Pdots@NH_2_ (Scale = 100 nm). (**B**) Particle size distribution of Pdots, Pdots@SH, Pdots@COOH, and Pdots@NH_2_ measured from DLS. (**C**) UV-vis-NIR absorption spectra of Pdots, Pdots@SH, Pdots@COOH, and Pdots@NH_2_. (**D**) Fluorescence spectra of Pdots, Pdots@SH, Pdots@COOH, and Pdots@NH_2_. (**E**) Zeta potential of Pdots, Pdots@SH, Pdots@COOH, and Pdots@NH_2_. (**F**) Polymer dispersity index (PDI) of Pdots, Pdots@SH, Pdots@COOH, and Pdots@NH_2_. (**G**) PDI of Pdots, Pdots@SH, Pdots@COOH, and Pdots@NH_2_ at different times (*n* = 3).

**Figure 3 molecules-28-02034-f003:**
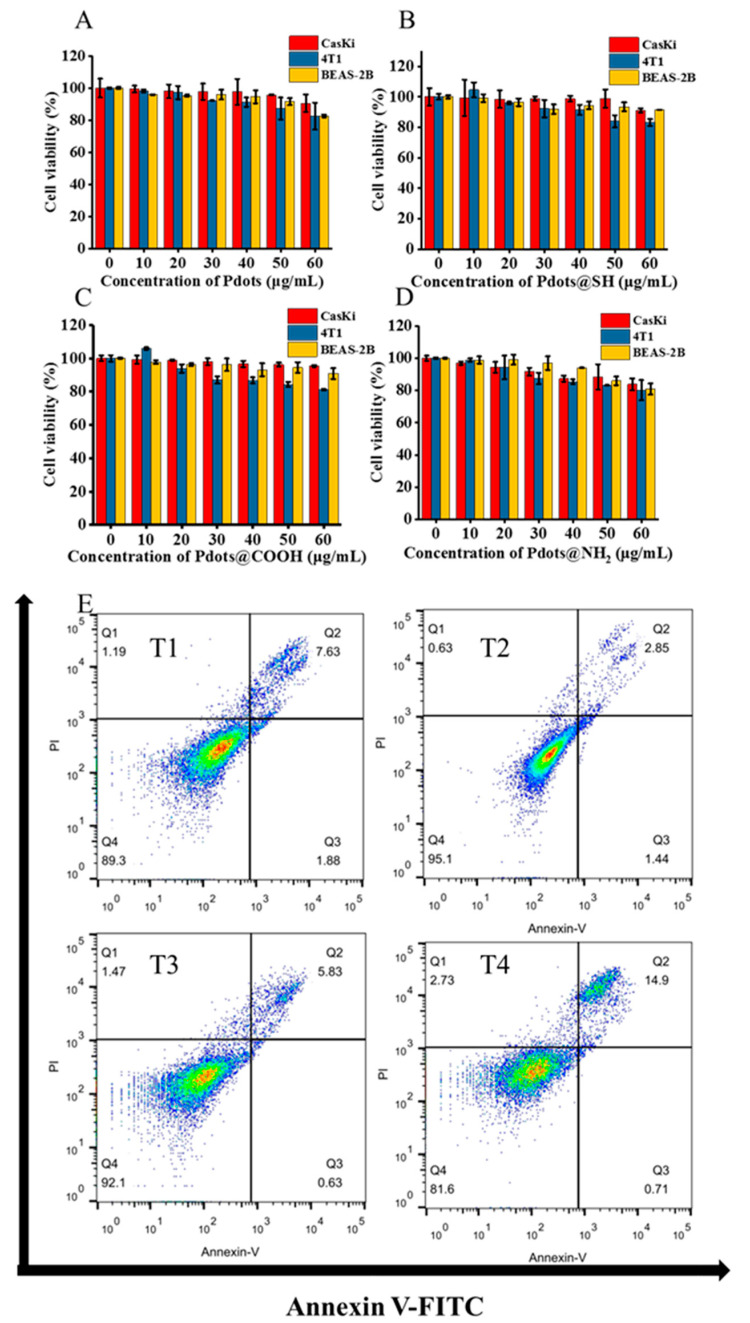
In vitro cytotoxicity study of Pdots, Pdots@SH, Pdots@COOH, and Pdots@NH_2_. Relative cell viability of CaSki, 4T1, and BEAS-2B cells after incubation with Pdots (**A**), Pdots@SH (**B**), Pdots@COOH (**C**), and Pdots@NH_2_ (**D**) at different concentrations (0, 1, 10, 20, 30, 40, 50, and 60 μg/mL) for 24 h. Data represent the mean ± SD (*n* = 5). (**E**) Flow cytometry analysis of the apoptosis of CaSki cells after different treatments. Groups T1–T4: T1: Pdots; T2: Pdots@SH; T3: Pdots@COOH; T4: Pdots@NH_2_ ([Pdots] = 50 μg/mL, [Pdots@SH] = 50 μg/mL, [Pdots@COOH] = 50 μg/mL, [Pdots@NH_2_] = 50 μg/mL).

**Figure 4 molecules-28-02034-f004:**
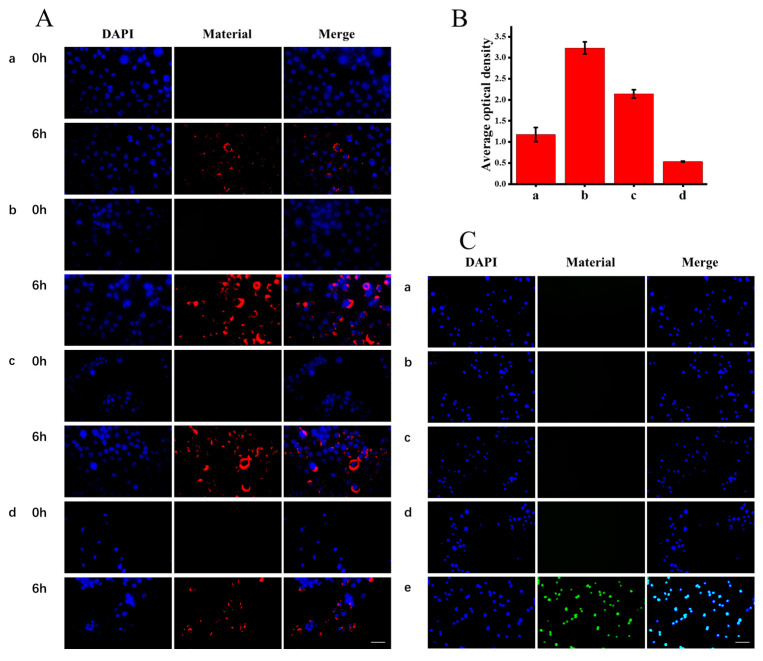
(**A**) Fluorescence images of CaSki cells after incubation with Pdots (a), Pdots@SH (b), Pdots@COOH (c), and Pdots@NH_2_ (d) ([Pdots] = 50 μg/mL, [Pdots@SH] = 50 μg/mL, [Pdots@COOH] = 50 μg/mL, [Pdots@NH_2_] = 50 μg/mL) for 0 and 6 h. The red fluorescence indicated the presence of materials, and cell nuclei (blue) were stained with 4′,6-diamidino-2-phenylindole (DAPI), scale bar = 50 μm. (**B**) Statistical analysis of fluorescence intensity of cells treated with Pdots (a), Pdots@SH (b), Pdots@COOH (c), and Pdots@NH_2_ (d) for 6 h according to A. Error bars represent the standard deviations (*n* = 3). (**C**) Confocal fluorescence images of CaSki cells stained by DCFH-DA (green) and DAPI (blue). 2,7-dichlorodi-hydrofluorescein diacetate (DCFH-DA) for ROS detection. (a): Pdots, (b): Pdots@SH, (c): Pdots@COOH, (d): Pdots@NH_2_, (e): positive control.

**Figure 5 molecules-28-02034-f005:**
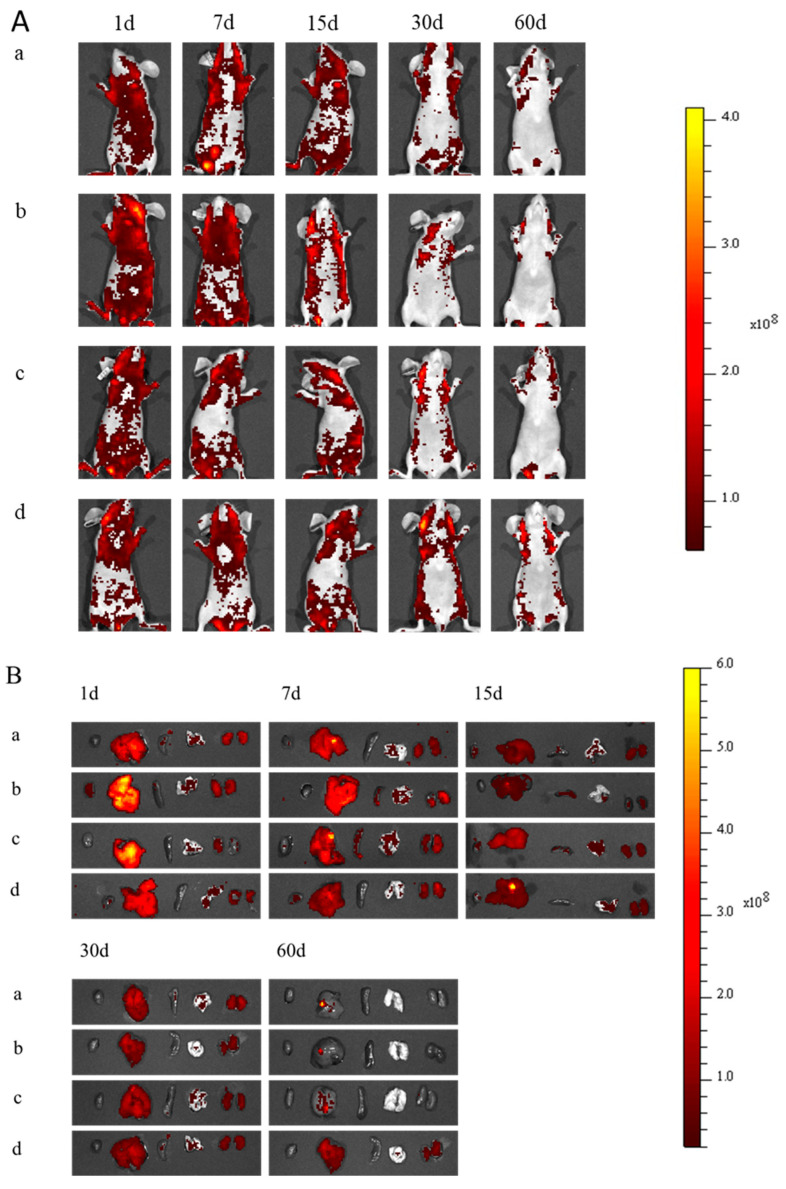
(**A**) Fluorescence images of Pdots (a), Pdots@SH (b), Pdots@COOH (c), and Pdots@NH_2_ (d). The nanoparticles (10 mg/kg) were injected into Balb/c mice via tail vein at different time points (1 day, 7 days, 15 days, 30 days, and 60 days; scale range 1.0–4.0 × 10^8^). (**B**) Fluorescence images of major organs (from left to right: heart, liver, spleen, lung, and kidney) of Balb/c mice at 1, 7, 15, 30, and 60 days after treatment with Pdots (a), Pdots@SH (b), Pdots@COOH (c), and Pdots@NH_2_ (d) (scale range 1.0–6.0 × 10^8^).

**Figure 6 molecules-28-02034-f006:**
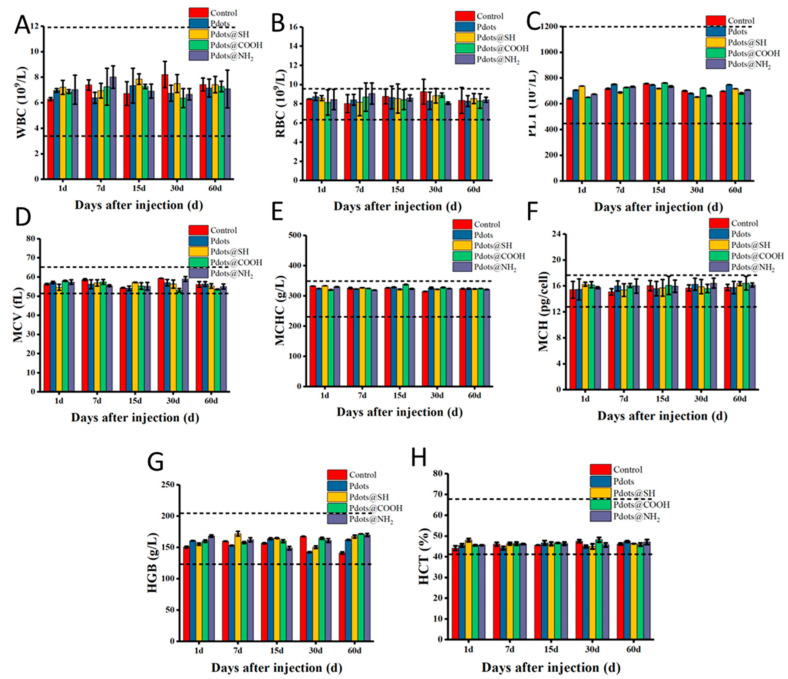
Complete blood counts of Balb/c mice treated with Pdots, Pdots@SH, Pdots@COOH, and Pdots@NH_2_ at 1 day, 7 days, 15 days, 30 days, and 60 days. (**A**) White blood cell (WBC), (**B**) red blood cell (RBC), and (**C**) platelet (PLT) counts, (**D**) mean corpuscular volume (MCV), (**E**) mean corpuscular hemoglobin concentration (MCHC), (**F**) mean corpuscular hemoglobin (MCH), (**G**) hemoglobin concentration (HGB), (**H**) hematocrit (HCT) of Balb/c mice. No statistically significant changes were observed between groups. The dashed line area in the figure shows the normal reference range for hematological data in healthy BALB/c mice.

**Figure 7 molecules-28-02034-f007:**
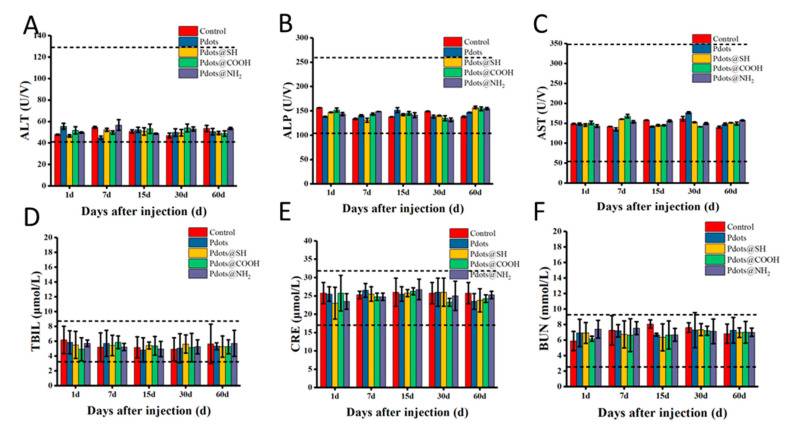
The serum biochemical analysis of Balb/c mice treated with Pdots, Pdots@SH, Pdots@COOH, and Pdots@NH_2_ at 1 day, 7 days, 15 days, 30 days, and 60 days. (**A**) Alkaline phosphatase (ALP), (**B**) alanine aminotransferase (ALT), (**C**) aspartate aminotransferase (AST), (**D**) blood urea nitrogen (BUN), (**E**) creatinine (CRE), (**F**) total bilirubin (TBIL). No statistically significant changes were observed between groups. The dashed line area in the figure shows the normal reference range for hematological data in healthy BALB/c mice.

**Figure 8 molecules-28-02034-f008:**
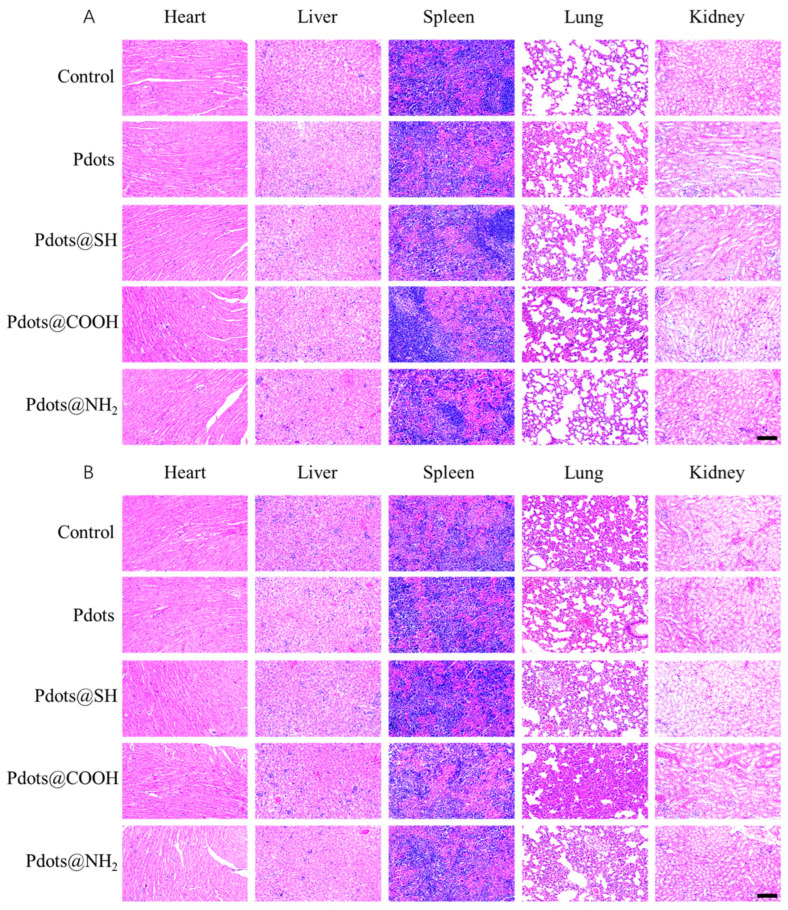
Hematoxylin- and eosin-stained images of major organs including the heart, liver, spleen, lungs, and kidneys from the Balb/c mice injected with Pdots, Pdots@SH, Pdots@COOH, and Pdots@NH_2_ at 1 day (**A**) and 60 days (**B**) post-injection, scale bar = 100 μm.

## Data Availability

Not applicable.

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
