# Peer review of "Biocompatibility and Biological Effects of Surface-Modified Conjugated Polymer Nanoparticles"

_molecules, 2023, doi:10.3390/molecules28052034_

Round 1

Reviewer 1 Report

In this manuscript, the authors investigated biocompatibility and biological effects of surface-modified conjugated polymer nanoparticles. The manuscript is well-written and results were discussed in detail. This research is helpful for the application of conjugated polymer nanoparticles in biomedical field. I would recommend the publication of the current work after addressing the following concerns:

1. The language needs some polishing prior to publication, such as naming of different modified polymer nanoparticles.

2. The experimental study for Histopathological Examination should be given in more detail?

3. Most of the references are older. Some relevant reference documentations introducing different types of nanoparticles should be supplemented, such as Adv. Mater 15 (2022) e2110364; ACS. Nano 5 (2021) 9101-9110; Angew. Chem. Int. Ed. 15 (2022) e202117433 and so on.

Author Response

Response to Reviewer 1: In this manuscript, the authors investigated biocompatibility and biological effects of surface-modified conjugated polymer nanoparticles. The manuscript is well-written and results were discussed in detail. This research is helpful for the application of conjugated polymer nanoparticles in biomedical field. I would recommend the publication of the current work after addressing the following concerns:

Detailed comments:

Comment 1: The language needs some polishing prior to publication, such as naming of different modified polymer nanoparticles.

Response 1: Thank you for your helpful advice. We have reviewed the article carefully and made changes throughout the article.

Comment 2: The experimental study for Histopathological Examination should be given in more detail?

Response 2: Thank you for your kind comments. In this experiment, H&E staining was performed on the major organs (heart, liver, spleen, lung and kidney), and the experimental results showed that no lesions were found in the organs and tissues, and all were normal in shape. We have added the H&E staining experiment details as Section 2.13.

Comment 3: Most of the references are older. Some relevant reference documentations introducing different types of nanoparticles should be supplemented, such as Adv. Mater 15 (2022) e2110364; ACS. Nano 5 (2021) 9101-9110; Angew. Chem. Int. Ed. 15 (2022) e202117433 and so on.

Response 3: Thank you very much for your recommendation. We have carefully read and searched the related literature mentioned by the reviewer. We have cited those literatures in the “Introduction” section in the revised manuscript as Refs3, 4 and 6. We also updated the relevant literature on page 1 and 2.

Reviewer 2 Report

Title: Biocompatibility and Biological Effects of Surface-Modified 2 Conjugated Polymer Nanoparticles

In tthis study, authors systematically investigated the biological effects and biocompatibility of Pdots with different surface modifications and revealed the interactions between Pdots and organisms at the cellular and animal levels. The surface of Pdots were modified with different functional groups. Extracellular studies showed that the modification of sulfhydryl, carboxyl, and amino groups had no significant effect on the physicochemical properties of Pdots, except that the amino modification affected the stability of Pdots to a certain extent. At the cellular level, Pdots@NH2 reduced cellular uptake capacity and increased cytotoxicity due to their instability in solution. At the in vivo level, the body circulation and metabolic clearance of Pdots@SH and Pdots@COOH were superior to those of Pdots@NH2.

This paper is recommended to accept for publication upon addressing the following points.

·         The authors have functionalized Pdots with thiol, carboxyl, and amino groups. The structural and morphological features of the functionalized specimens:  Pdots Pdots@SH, Pdots@COOH and Pdots@NH2 were observed by electron microscopy technique (TEM). However, the functionalization and the morphology of the prepared materials is not clear by the provided TEM Fig. 2A. Authors should provide the clear and high-quality images of Pdots, Pdots@SH, Pdots@COOH and Pdots@NH2.

·         What are the image dimensions of Figure 5?  The text on Figure 5A is not readable.

·         Authors should provide the number of readings for biochemical analysis (Figs. 6 & 7).

·         Authors should mention the magnifications and add scale bars for Figure 8. Also, indicate or mark the important findings if applicable on such photographs.

·         Authors concluded the biocompatibility and biodistribution of thiol-, carboxyl-, and amino-modified Pdots at the cellular and living animal levels but this claim is not clear. Authors should explain and highlight this point further.

·         Figure 7: The morphology of the cells is not clear when observed only with confocal microscopy. Authors should provide the scanning electron microscopy (SEM) examination of control, treated cells with individual and AgNPs-ZnPs system in order to see their effects.

Author Response

Response to Reviewer 2: In this study, authors systematically investigated the biological effects and biocompatibility of Pdots with different surface modifications and revealed the interactions between Pdots and organisms at the cellular and animal levels. The surface of Pdots were modified with different functional groups. Extracellular studies showed that the modification of sulfhydryl, carboxyl, and amino groups had no significant effect on the physicochemical properties of Pdots, except that the amino modification affected the stability of Pdots to a certain extent. At the cellular level, Pdots@NH2 reduced cellular uptake capacity and increased cytotoxicity due to their instability in solution. At the in vivo level, the body circulation and metabolic clearance of Pdots@SH and Pdots@COOH were superior to those of Pdots@NH2. This paper is recommended to accept for publication upon addressing the following points.

Comment 1: The authors have functionalized Pdots with thiol, carboxyl, and amino groups. The structural and morphological features of the functionalized specimens:  Pdots Pdots@SH, Pdots@COOH and Pdots@NH2 were observed by electron microscopy technique (TEM). However, the functionalization and the morphology of the prepared materials is not clear by the provided TEM Fig. 2A.  Authors should provide the clear and high-quality images of Pdots, Pdots@SH, Pdots@COOH and Pdots@NH2.

Response 1: Thank you for your kind comments. We have replaced figure 2A with TEM images with higher resolution.

Comment 2: What are the image dimensions of Figure 5?  The text on Figure 5A is not readable.

Response 2: Thank you very much for your wonderful suggestion. We have replaced the clearer image and indicated the scale range in figure 5A.

Comment 3: Authors should provide the number of readings for biochemical analysis (Figs. 6&7).

Response 3: Thank you very much for the comments. We have marked the dotted line area in Figs. 6&7 as the normal reference range for hematological data in healthy BALB/c mice.

Comment 4: Authors should mention the magnifications and add scale bars for Figure 8. Also, indicate or mark the important findings if applicable on such photographs.

Response 4: Thank you for your helpful suggestion. We added scale bar = 100μm in the interpretation section of figure 8.

Comment 5: Authors concluded the biocompatibility and biodistribution of thiol-, carboxyl-, and amino-modified Pdots at the cellular and living animal levels but this claim is not clear. Authors should explain and highlight this point further.

Response 5: Thank you for your helpful suggestion. We have revised our conclusions as the following: we investigated the biocompatibility and biodistribution of thiol-, carboxyl-, and amino-modified Pdots at the cellular and living animal levels. In vitro studies showed that the cellular uptake of thiol and carboxyl groups modified Pdots was strong and had no significant impact on cell viability, apoptosis and oxidative stress. Amino modification decreased the uptake of Pdots cells and increased the cytotoxicity of Pdots to some extent. Next, we conducted an in vivo study and found that the nanoparticles were distributed in major tissues and organs of mice after injection, with the largest distribution in the liver. As time went by, the nanoparticles were eliminated from the mice by metabolism through the digestive system. Moreover, the in vivo circulation and metabolic clearance of Pdots@SH and Pdots@COOH were better than those of Pdots@NH2. The blood routine analysis, serum biochemical analysis and histopathological analysis showed that Pdots modified with different modifications had no obvious toxicity and did not affect liver and kidney functions. The study demonstrated that thiol and carboxyl modified Pdots have better biocompatibility than amino modified Pdots, which lays a solid foundation for the biomedical application and clinical translation of Pdots.

Comment 6: Figure 7: The morphology of the cells is not clear when observed only with confocal microscopy. Authors should provide the scanning electron microscopy (SEM) examination of control, treated cells with individual and AgNPs-ZnPs system in order to see their effects.

Response 6: We want to appreciate the reviewer’s comments for improving our manuscript. However, we believe this comment might not belong to our manuscript. Here is the reason: Our Figure 7 in the original manuscript is the biochemical analysis, but no morphology of the cells with confocal microscopy. Also, we were not investigating AgNPS-ZnPs system in our manuscript at all. Please double check and confirm with the reviewer. Thanks for the consideration.